# A covariation analysis reveals elements of selectivity in quorum sensing systems

Samantha Wellington Miranda[1], Qian Cong[2,3,4,5], Amy L Schaefer[1],
Emily Kenna MacLeod[1], Angelina Zimenko[1], David Baker[2,3,6],
E Peter Greenberg[1]*

[1]Department of Microbiology, University of Washington, Seattle, United States;
[2]Department of Biochemistry, University of Washington, Seattle, United States;
[3]Institute for Protein Design, University of Washington, Seattle, United States;
[4]Eugene McDermott Center for Human Growth and Development, University of
Texas Southwestern Medical Center, Dallas, United States; [5]Department of
Biophysics, University of Texas Southwestern Medical Center, Dallas, United States;
[6]Howard Hughes Medical Institute, University of Washington, Seattle, United States

**Abstract** Many bacteria communicate with kin and coordinate group behaviors through a form of cell-cell signaling called acyl-homoserine lactone (AHL) quorum sensing (QS). In these systems, a signal synthase produces an AHL to which its paired receptor selectively responds. Selectivity is fundamental to cell signaling. Despite its importance, it has been challenging to determine how this selectivity is achieved and how AHL QS systems evolve and diversify. We hypothesized that we could use covariation within the protein sequences of AHL synthases and receptors to identify selectivity residues. We began by identifying about 6000 unique synthase-receptor pairs. We then used the protein sequences of these pairs to identify covariation patterns and mapped the patterns onto the LasI/R system from *Pseudomonas aeruginosa* PAO1. The covarying residues in both proteins cluster around the ligand-binding sites. We demonstrate that these residues are involved in system selectivity toward the cognate signal and go on to engineer the Las system to both produce and respond to an alternate AHL signal. We have thus demonstrated that covariation methods provide a powerful approach for investigating selectivity in protein-small molecule interactions and have deepened our understanding of how communication systems evolve and diversify.

*For correspondence:
epgreen@u.washington.edu

Competing interest: See
page 17

Reviewing editor: Michael T
Laub, Massachusetts Institute of
Technology, United States

## Introduction

Quorum sensing (QS) is a widespread form of cell-cell signaling that bacteria use to coordinate the production of public goods including toxins, antibiotics, bioluminescence, and secreted enzymes (*Waters and Bassler, 2005*; *Whiteley et al., 2017*). Many Proteobacteria (*Case et al., 2008*) and Nitrospirae (*Mellbye et al., 2017*) employ a form of QS based on acyl-homoserine lactone (AHL) signals. AHL QS systems consist of two proteins: a LuxI-type signal synthase and a LuxR-type receptor (*Figure 1A*). The signal synthase produces an AHL from *S*-adenosyl methionine (SAM) and an acyl-acyl carrier protein for some LuxI-type synthases or an acyl-coenzyme A substrate for others (*Schaefer et al., 2008*; *Figure 1B*). AHL signals can freely diffuse through cell membranes (*Kaplan and Greenberg, 1985*; *Pearson et al., 1999*) and at low cell density the QS system is 'off'. At high cell density, the signal accumulates and binds the LuxR-type receptor which is a cytosolic transcription factor that regulates gene expression in response to signal binding.

AHL signals share a conserved lactone core, but vary in the acyl moiety which can be a fatty acid ranging from 4 to 20 carbons long, with potential oxidation on the C3 carbon and varying degrees

**eLife digest** Communication is vital in any community and it is no different for bacteria. Some of the microbes living in bacterial communities are closely related to one another and can help each other survive and grow. They do this by releasing chemical signals that coordinate their behaviors, including those that are damaging to the infected host.

A key aspect of this coordination is knowing how many related bacteria are present in a given environment. In a process known as quorum sensing, the bacteria release a chemical signal which neighboring sibling bacteria detect and respond to. The larger the population of bacteria, the more the signal accumulates. At a certain threshold, the signal activates the genes needed to trigger a coordinated action, such as producing toxins or antibiotics. Many bacteria communicate using acylhomoserine lactone signaling systems, which involve different signals depending on the species of bacteria. But it is unclear how this diversity evolved, and how bacteria can distinguish between signals from related and unrelated bacterial cells.

To understand this, Wellington Miranda et al. used computational techniques to analyze how the proteins responsible for acylhomoserine lactone signaling coevolved. The analysis identified specific parts of these proteins that determine which signal will be produced and which will trigger a bacterium into action. Wellington Miranda et al. then used these insights to engineer the bacteria *Pseudomonas aeruginosa* to produce and respond to a signal that is naturally made by another bacterial species.

These computational methods could be used to analyze other proteins that have coevolved but do not physically interact. Within the area of quorum sensing, this approach will help to better understand the costs and benefits of signal selectivity. This may help to predict bacterial interactions and therefore behaviors during infections.

of unsaturation, or can have an aromatic or branched structure (*Rajput et al., 2016*). This variability in the acyl portion of the signal confers selectivity to the system. Typically, a LuxI-type synthase produces a primary AHL to which its paired LuxR-type receptor selectively responds (*Aframian and Eldar, 2020*). Selectivity is critical to cell signaling in order to avoid undesired cross-talk or spurious outputs (*Laub, 2016*). In the case of QS, selectivity ensures bacteria cooperate only with kin cells.

Despite its importance, we know little about how QS systems achieve selectivity or how they evolve and diversify to use new signals. Although the conserved amino acids essential for synthase and receptor activity are well described (*Parsek et al., 1997*; *Zhang et al., 2002*), residues that dictate selectivity are often different from those that are required for activity (*Collins et al., 2005*). Due to the low amino acid sequence identity between LuxI/R homologues, it has been difficult to determine how QS systems discriminate between various AHL signals (*Fuqua et al., 1996*).

We hypothesized that we could use covariation patterns to identify QS selectivity residues. Such methods have been used to identify amino acid residues that interact with each other within proteins and between proteins that physically bind one another (*Aakre et al., 2015*; *Ovchinnikov et al., 2014*; *Skerker et al., 2008*). Here, we endeavored to expand these methods to assess the interaction between AHL synthases and receptors. While AHL synthases and receptors do not physically interact, they interact indirectly via binding to a shared cognate signal and are believed to coevolve to maintain this shared signal recognition (*Aframian and Eldar, 2020*). Phylogenetic analyses also support coevolution of synthases and receptors (*Gray and Garey, 2001*; *Lerat and Moran, 2004*). We therefore hypothesized that we could apply covariation methods in a novel way to identify amino acid residues that covary between QS synthases and receptors, and further, that the covarying residues would be those responsible for signal selectivity.

We used a statistical method, GREMLIN (*Kamisetty et al., 2013*), to measure covariation within the sequences of AHL synthase-receptor pairs and mapped the covarying residues onto the LasI/R QS system of *Pseudomonas aeruginosa* PAO1. By engineering substitutions in the top-scoring residues identified by GREMLIN, we demonstrate that they are indeed important for signal selectivity and, further, that these residues can be used to rationally engineer LasI/R to produce and respond to a non-native signal. We thus provide a proof of principle for a new use of covariation methods to

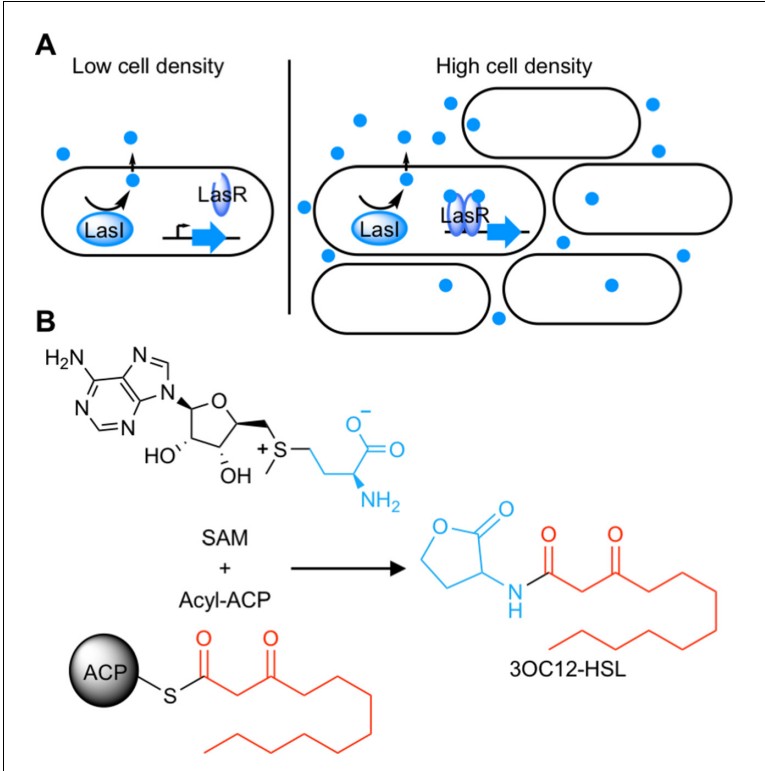

**Figure 1.** Schematic of acyl-homoserine lactone (AHL) quorum sensing (QS). (**A**) AHL QS circuits contain a signal synthase, a LuxI homolog (LasI in this cartoon), that produces an AHL signal. At low cell densities, the system exists in an 'off' state. At high cell densities, the AHL concentration increases and the signal binds its cytosolic receptor, a LuxR homolog (LasR in this cartoon), which functions as a transcription factor. (**B**) LuxI-type synthases produce AHL signals from two substrates: *S*-adenosyl methionine (SAM) and an activated organic acid in the form of an acyl-acyl carrier protein (ACP) or, in some cases, an acyl-coenzyme A (CoA). SAM provides the lactone core, which is conserved across all AHL signals, while the acyl-ACP (shown here) provides an acyl moiety which varies between signaling systems. In this example, the synthase LasI produces *N*-3-oxo-dodecanoyl-L-homoserine lactone (3OC12-HSL).

investigate selectivity in non-physical protein-protein interactions and at the same time identify determinants of QS signal selectivity.

## Results

### Covariation patterns in QS systems

To begin our analysis, we gathered select protein sequences for known synthase-receptor pairs (*Supplementary file 1A*) and used these sequences to search the European Nucleotide Archive (ENA) database (*Amid et al., 2019*) from the European Bioinformatics Institute and the Integrated Microbial Genomes and Microbiomes (IMG/M) database (*Chen et al., 2021*) from the Joint Genome Institute (JGI) for additional synthase-receptor pairs. The genes for synthase-receptor pairs are frequently co-located on the genome, and organisms can harbor more than one complete QS system (*Fuqua et al., 1996*). To increase the likelihood of identifying true pairs, we required that the two genes be separated by no more than two coding sequences. A total of 6360 non-identical pairs were identified. We further discarded pairs that were more than 90% identical to another pair, resulting in 3489 representative AHL synthase-receptor pairs.

We aligned these sequences to the LasI/R QS system from *P. aeruginosa* PAO1. Not only is *P. aeruginosa* a clinically important pathogen, the Las system is well studied and crystal structures have been solved for both LasI (*Gould et al., 2004*) and LasR (*Zou and Nair, 2009*), making this a particularly useful model system for our studies. We connected the sequences of the synthase and the

receptor from each pair and used GREMLIN to analyze covariation patterns in these sequences. We applied average product correction (APC) to the GREMLIN covariance coefficients, a common technique shown to improve the accuracy of coevolution analyses (*Buslje et al., 2009*). An overview of our workflow is shown in *Figure 2—figure supplement 1*. We performed the same analysis by aligning the synthase-receptor pairs to the LuxI/R system from *Vibrio fischeri* MJ11. The top-ranking coevolving residue pairs overlap significantly between the LasI/R and LuxI/R systems (62.5% in common among the top 0.05% residue pairs) (*Figure 2—figure supplement 2A–D*). We integrated the analyses for the LasI/R and LuxI/R systems by using the higher score for each residue pair and the top 10 residue pairs are shown in *Figure 2A* and *Figure 2—figure supplement 3A*. As a control, we randomly paired the synthases and receptors from different species and reanalyzed them using GREMLIN. While top-scoring covarying residues had a minimal GREMLIN score (with APC) of 0.09, the highest score from the randomized control was 0.08 (*Figure 2B* and *Figure 2—figure supplement 3B*). This control provides a guideline for our analysis; residues with a GREMLIN score (with APC) above or near the maximal score for the randomized control are likely to be meaningful.

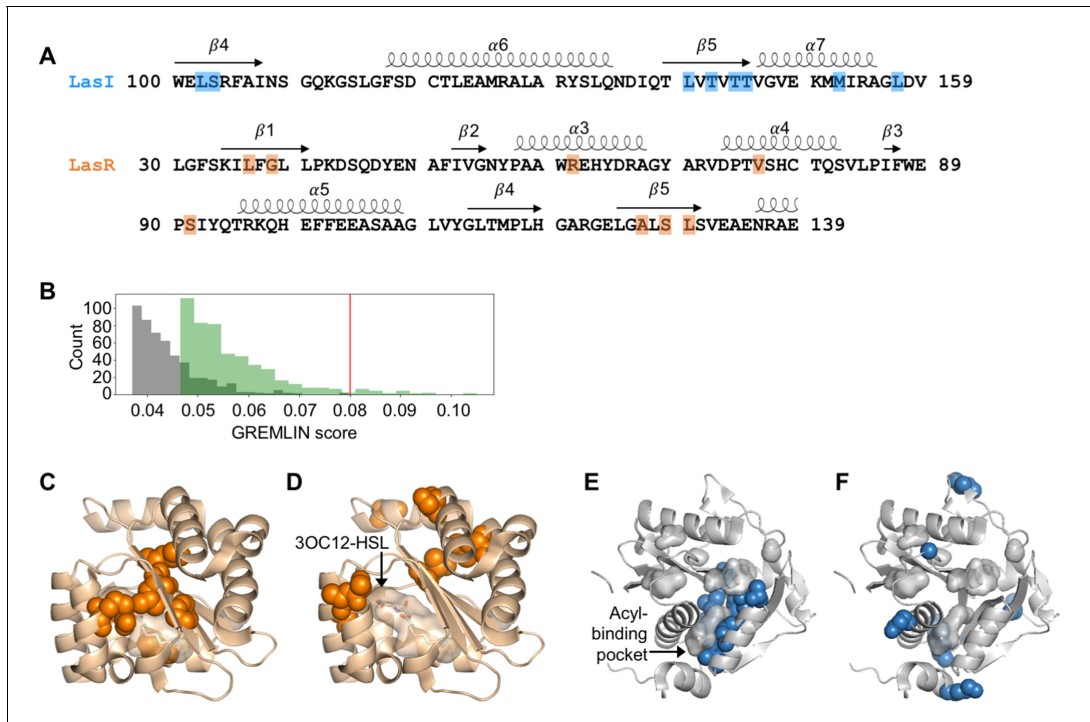

**Figure 2.** Covarying residues identified in LasI/R. (**A**) Top-scoring covarying residues based on an integrated analysis of the Las and Lux systems are shaded in blue or orange on the primary amino acid sequences of LasI (synthase) and LasR (receptor), respectively. Secondary structures, shown as arrows (sheets) and loops (helixes), are based on crystal structures for LasI and LasR (PDB 1RO5 and PDB 3IX3, respectively). The list of top-scoring residue pairs along with their GRELMIN scores (with average product correction [APC]) is shown in *Figure 2—figure supplement 3A*. (**B**) Distribution of GREMLIN scores (with APC) for the top 500 pairs from the Las analysis (green) and the randomized control (gray). The top score in the control was 0.08, as indicated by the red line. (**C**) Top-scoring covarying residues mapped onto the N-terminal LasR ligand-binding domain (covarying residues in orange, *N*-3-oxo-dodecanoyl-L-homoserine lactone [3OC12-HSL] colored by element; PDB 3IX3) and (**E**) LasI (covarying residues in blue; PDB 1RO5). Top-scoring residues in the randomized control are mapped onto (**D**) LasR and (**F**) LasI as in panels **C** and **E**.

The online version of this article includes the following source data and figure supplement(s) for figure 2:

**Source data 1.** Acyl-homoserine lactone (AHL) synthase and receptor sequences used in this study, aligned to LasI/R.

**Source data 2.** Acyl-homoserine lactone (AHL) synthase and receptor sequences used in this study, aligned to LuxI/R.

**Figure supplement 1.** Workflow for the identification of covarying residues in acyl-homoserine lactone (AHL) quorum sensing (QS) synthases and receptors.

**Figure supplement 2.** Top-scoring GREMLIN residues.

**Figure supplement 3.** Covarying residues identified in LasI/R and in a randomized control.

## Top-scoring residues cluster near ligand-binding pockets

For both LasI and LasR, the top-scoring covarying residues cluster around the ligand-binding pocket. For LasR, the top-scoring residues map exclusively to the ligand-binding domain with an average distance of 5.0 Å from the co-crystalized native ligand N-3-oxo-dodecanoyl-L-homoserine lactone (3OC12-HSL) (*Figure 2C*). In contrast, the top residues identified in the randomized control are scattered throughout LasR, including three residues in the DNA-binding domain, and are an average distance of 18 Å from 3OC12-HSL (*Figure 2D*).

In LasI, the top-scoring covarying residues cluster around the hydrophobic pocket thought to bind the fatty acyl substrate (*Figure 2E*) and are an average distance of 3.7 Å from an acyl substrate modeled into the LasI structure (*Gould et al., 2004*). As with LasR, the residues identified in the randomized control are scattered throughout LasI, with many of the residues exposed to solvent (*Figure 2F*). The randomized control residues in LasI are over three times further from the fatty acyl substrate, mean distance = 12 Å, compared to the covarying residues.

Due to their location near the ligand-binding pockets, several of the covarying residues have been previously studied in various LasI/R homologues. Encouragingly, many of these residues have been reported to be important for protein activity and, in some cases, for selectivity. We have summarized several of these studies in *Supplementary files 1B-C*.

## LasR substitutions alter selectivity

To determine whether residues identified by GREMLIN are involved in LasR selectivity, we introduced substitutions into a selection of the top-scoring amino acids, G38, R61, A127, S129, and L130, focusing on common natural variants at each position (*Supplementary file 1D*). By expressing LasR in *Escherichia coli* and measuring its activity against a previously reported panel of 19 AHL signals (*Figure 3—figure supplement 1*; *Wellington and Greenberg, 2019*), we were able to quickly prioritize variants for further study. The majority of our LasR variants retained the ability to respond to AHLs and had an altered selectivity profile when compared to wild type (*Figure 3*). For example, LasR$^{G38V}$ responds to N-3-oxo-tetradecanoyl-L-homoserine lactone (3OC14-HSL) and N-3-oxo-hexadecanoyl-L-homoserine lactone (3OC16-HSL) but not 3OC12-HSL, while LasR$^{L130F}$ is more subtly altered and responds more strongly than wild type to N-3-oxo-octanoyl-L-homoserine lactone (3OC8-HSL) and N-3-oxo-decanoyl-L-homoserine lactone (3OC10-HSL), consistent with a previous study of this amino acid substitution (*McCready et al., 2019*).

We, and others, have previously demonstrated that compared to native activity, QS receptor sensitivity and selectivity can be altered when the receptor gene is expressed in *E. coli* or overexpressed in *P. aeruginosa* (*Moore et al., 2015*; *Wellington and Greenberg, 2019*). We therefore engineered several mutations in *lasR* on the *P. aeruginosa* PAO-SC4 chromosome to study LasR activity in the native context. *P. aeruginosa* PAO-SC4 is an AHL synthase-null mutant which we use here to measure LasR activity in response to exogenously provided AHL signals. We measured LasR activity by using a transcriptional reporter in which the promoter of the LasR-regulated gene *rsaL* controls *gfp* expression (*Wellington and Greenberg, 2019*). This provides a direct measure of LasR activity as a transcriptional activator.

The *lasR* mutations largely had the same effect on activity and selectivity in *P. aeruginosa* as they did when *lasR* was expressed in *E. coli* (*Figure 4—figure supplement 1A*). Consistent with its role in forming a water-mediated hydrogen bond with the C3 oxygen of 3OC12-HSL, and with previous studies (*Collins et al., 2006*; *Gerdt et al., 2015*), LasR R61 mutants were less responsive to oxo-substituted AHLs, but maintained wild type or better levels of activation by unsubstituted AHLs (*Figure 4* and *Figure 4—figure supplement 1A*). On the other hand, LasR$^{A127L}$ had an increased sensitivity to numerous signals (*Figure 4*). LasR$^{A127M}$ was also more strongly activated than wild type by multiple signals, as was LasR$^{L130F}$ (*Figure 4—figure supplement 1A*). Residue A127 interacts with 3OC12-HSL near the middle of its fatty acyl tail. The A127M and A127L substitutions may increase signal binding through increased hydrophobic contacts with the acyl chain. Residue L130 is near the homoserine lactone core of the bound signal. The L130F substitution results in structural changes that potentially increase LasR stability, thereby broadening its selectivity (*McCready et al., 2019*).

The amino acid substitutions also affect the sensitivity of LasR to 3OC12-HSL (*Figure 4E* and *Figure 4—figure supplement 1B*). Interestingly, two of our variants are more sensitive to 3OC12-HSL than wild-type LasR. LasR$^{A127L}$ is roughly threefold more sensitive and LasR$^{L130F}$ is twofold more

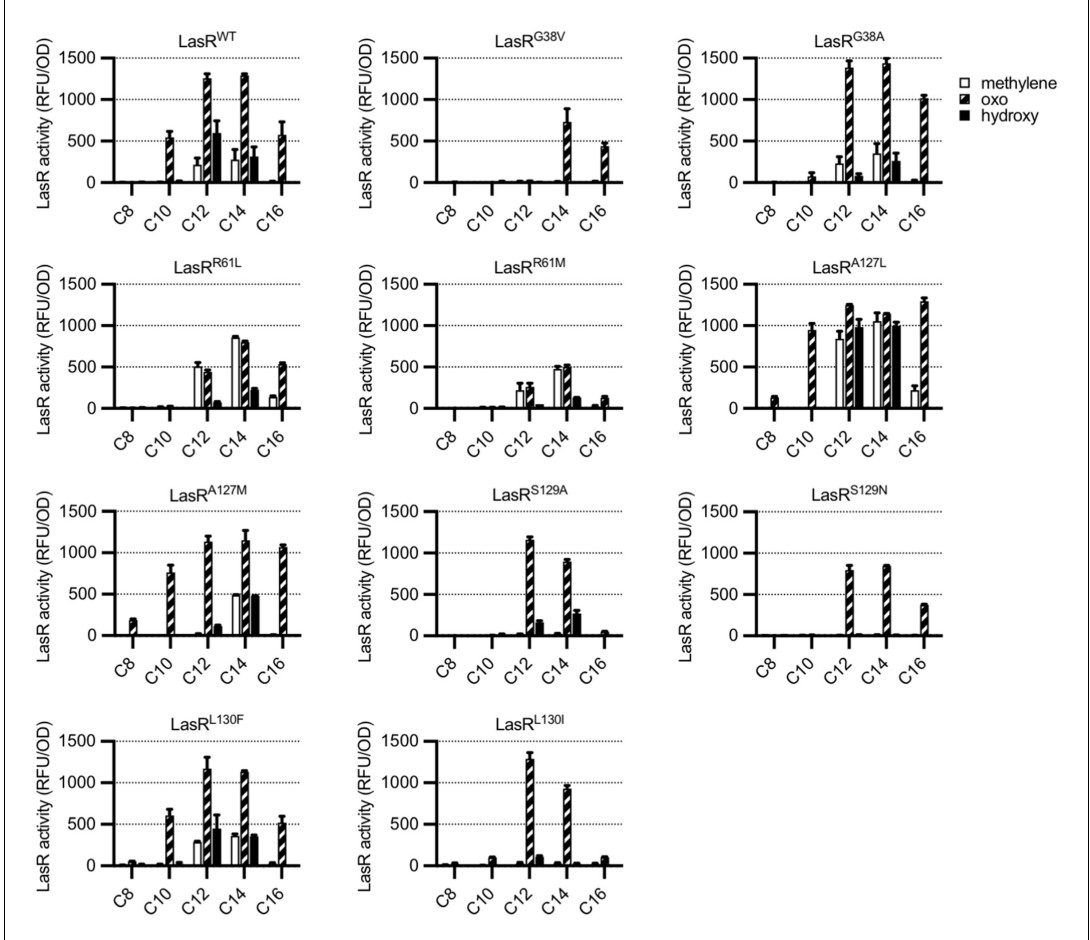

**Figure 3.** Activity of LasR variants in *Escherichia coli*. LasR activity in response to a panel of acyl-homoserine lactone (AHL) signals (100 nM; chain length indicated on horizontal axis, C3 modification indicated by shading, structures and names are shown in *Figure 3—figure supplement 1*) is reported as relative fluorescence units (RFU) normalized by optical density at 600 nm (OD). Wild-type *lasR* (WT) or *lasR* with the indicated amino acid substitution were expressed from pJNL in *E. coli* harboring pPROBE-P$_{rsaL}$. Signals with four or six carbons in the acyl chain did not activate any of the LasR variants and are not shown. The following LasR variants had little or no activity in response to 100 nM AHLs: R61V, R61Q, S129T, G38L. Data are the mean and standard deviation of two biological replicates and are representative of three (variants) or ten (WT) independent experiments.

The online version of this article includes the following figure supplement(s) for figure 3:

**Figure supplement 1.** Acyl-homoserine lactone (AHL) panel used in this study.

sensitive. This increased sensitivity comes at the cost of decreased selectivity for both of these mutant proteins. In fact, many of our single amino acid variants displayed reduced selectivity compared to wild-type LasR (*Figure 3* and *Figure 4—figure supplement 1A*). The observed changes in LasR sensitivity and selectivity could be due to multiple factors including altered receptor affinity for a signal, altered protein stability, or both. The stability of LuxR-type receptors is intertwined with signal binding. QS systems are subject to elaborate control, a key component of which is that QS receptors, including LasR, are unstable and insoluble in the absence of bound signal (*Oinuma and Greenberg, 2011*; *Sappington et al., 2011*). Thus, changes in signal affinity usually lead to changes in the amount of soluble receptor present in a cell (*McCready et al., 2019*). Conversely, changes to the expression level of the receptor can lead to altered signal sensitivity and selectivity (*Wellington and Greenberg, 2019*). To determine whether receptor stability was affected in our variants, we assessed the abundance of soluble LasR in *P. aeruginosa* by immunoblotting in a selection of variants with altered 3OC12-HSL sensitivity: LasR$^{A127L}$, LasR$^{L130F}$, and LasR$^{R61L}$. Comparing these variants to wild type, there were not substantial changes in the abundance of soluble LasR, though LasR$^{R61L}$ was somewhat less abundant than wild type (*Figure 4—figure supplement 2*).

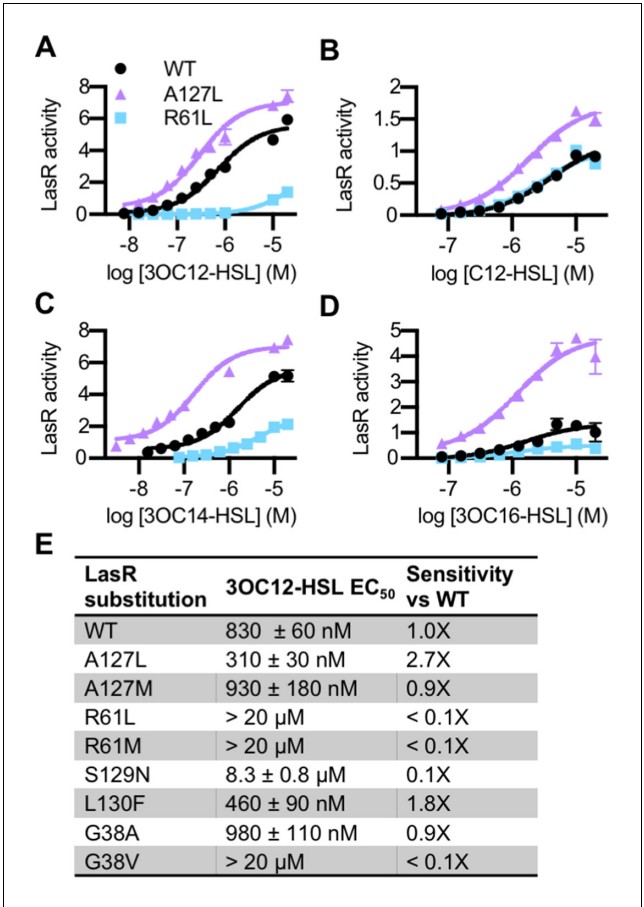

**Figure 4.** Activity of LasR variants in *Pseudomonas aeruginosa*. Activity of chromosomal *lasR* mutants in *P. aeruginosa* PAO-SC4 pPROBE-P$_{rsaL}$ in response to (**A**) *N*-3-oxo-dodecanoyl-L-homoserine lactone (3OC12-HSL), (**B**) *N*-dodecanoyl-L-homoesrine lactone (C12-HSL), (**C**) *N*-3-oxo-tetradecanoyl-L-homoserine lactone (3OC14-HSL), or (**D**) *N*-3-oxo-hexadecanoyl-L-homoserine lactone (3OC16-HSL). Structures of the signals are shown in *Figure 3—figure supplement 1*. Amino acid substitutions are indicated. Wild type (WT) is shown in black, LasR$^{A127L}$ in purple, and LasR$^{R61L}$ in blue. The horizontal axis indicates acyl-homoserine lactone (AHL) concentration. LasR activity is reported on the vertical axis as relative fluorescence units normalized by optical density at 600 nm (RFU/OD × 1000). Data are the mean and standard deviation of three biological replicates and are representative of three independent experiments. (**E**) The half maximal effective concentration (EC$_{50}$) of 3OC12-HSL for *P. aeruginosa* PAO-SC4 LasR variants. Data are the mean and SEM of three (variants) or six (WT) independent experiments (representative data shown in *Figure 4—figure supplement 1B*). Sensitivity of the variants compared to LasR$^{WT}$ is calculated by dividing WT EC$_{50}$ by variant EC$_{50}$.

The online version of this article includes the following source data and figure supplement(s) for figure 4:

**Figure supplement 1.** Activity of LasR variants in *Pseudomonas aeruginosa*.

**Figure supplement 2.** LasR solubility.

**Figure supplement 2—source data 1.** Chemiluminescent anti-LasR immunoblot image shown in *Figure 4—figure supplement 2A*.

**Figure supplement 2—source data 2.** Chemiluminescent anti-LasR immunoblot image shown in *Figure 4—figure supplement 2A* with lanes and bands labeled.

**Figure supplement 2—source data 3.** Chemiluminescent anti-LasR immunoblot image shown in *Figure 4—figure supplement 2B*.

**Figure supplement 2—source data 4.** Chemiluminescent anti-LasR immunoblot image shown in *Figure 4—figure supplement 2B* with lanes and bands labeled.

## LasI substitutions alter activity and selectivity

Similar to LasR, we focused our LasI amino acid alterations on the top-scoring positions: L102, T142, T145, and L157 (*Supplementary file 1E*). We expressed wild-type or mutated *lasI* on a low copy number plasmid in the AHL synthase-null *P. aeruginosa* PAO-SC4 and extracted AHLs produced by these bacteria from culture fluid. While bioassays are commonly used for the detection of AHLs (*Chu et al., 2011*), they suffer from multiple drawbacks. In particular, bioassays are not equally sensitive to all AHLs and typically cannot be used to determine which AHLs are produced and in what ratio. To screen our LasI variants for altered activity and selectivity, we developed a thin layer chromatography (TLC) method based on our existing high-performance liquid chromatography (HPLC) radiotracer assay (*Schaefer et al., 2018*). In this method, the C1 position in the homoserine lactone ring is labeled with $^{14}C$. The label is incorporated into AHLs at a ratio of one $^{14}C$ per AHL molecule. This results in unbiased detection of all AHLs produced. While the established method uses HPLC to separate and detect AHLs one sample at a time, we can run nine samples per TLC plate, resulting in a more high-throughput assay.

Using our TLC method, we confirmed that *lasI* directs the synthesis of the same primary product whether it is expressed on a plasmid or from the chromosome (*Figure 5—figure supplement 1A*). HPLC analysis of matched extracts confirmed that the major LasI product observed by TLC is 3OC12-HSL (*Figure 5—figure supplement 1B*). As expected, an empty vector control did not produce detectable AHLs, nor did we detect radioactivity in a media-only control. We then screened the activity of each *lasI* mutant by TLC (*Figure 5—figure supplement 1C–E*). Several mutants produced little or no detectable AHLs, but others, such as LasI$^{T145S}$, appeared to produce more 3OC12-HSL than wild-type LasI. These changes in selectivity and rate of synthesis may be due to biochemical changes in LasI or to altered protein expression or stability. We analyzed select extracts by both TLC and HPLC and found that the results were consistent between the two methods, further validating the TLC method.

Based on our TLC results, we selected one variant with altered selectivity, LasI$^{L157W}$, for further study by HPLC. By TLC, LasI$^{L157W}$ produces three signals: 3OC12-HSL and two signals of unknown identity (*Figure 5—figure supplement 1D*). Using HPLC, we found that LasI$^{L157W}$ produces equal amounts of two $^{14}C$-AHLs consistent with 3OC10-HSL and 3OC8-HSL along with a lesser amount of 3OC12-HSL (*Figure 5*). Residue L157 is located near the bottom of the acyl-binding pocket where it likely interacts with the end of the 12-carbon acyl chain in 3OC12-HSL. The L157W substitution could decrease the volume of the pocket, improving affinity for shorter substrates. These findings demonstrate that the covarying residues identified by GREMLIN influence LasI activity and selectivity, and that a single amino acid substitution is sufficient to significantly alter LasI selectivity.

## Covarying residues facilitate rational engineering of LasI/R selectivity

In general, multiple amino acid changes are required to generate a protein with orthogonal selectivity (*Aakre et al., 2015*; *Collins et al., 2006*; *Skerker et al., 2008*). In non-QS proteins, altered selectivity has been engineered by swapping the covarying residues in one homolog to the identities in another (*Aakre et al., 2015*; *Skerker et al., 2008*). Here, we seek to 'rewire' LasI/R to use an orthogonal signal. We targeted the MupI/R system from *Pseudomonas fluorescens* NCIMB 10586, which uses the signal 3OC10-HSL (*Hothersall et al., 2011*). MupI and MupR share 52% and 39% identity with LasI and LasR, respectively. MupI/R is among the systems closest in sequence identity to LasI/R that use a signal other than 3OC12-HSL.

LasR and MupR differ at eight covariation sites in the ligand-binding domain with a GREMLIN score (with APC) > 0.08 (*Figure 6—figure supplement 1A*). LasR modified to contain all eight substitutions was inactive. However, there were several intermediate variants that displayed an increased response to 3OC10-HSL. We identified three amino acid substitutions that are sufficient for this increased sensitivity: L125F, A127M, and L130F (*Figure 6A*, *Figure 6—figure supplement 1B*). LasR$^{L125F, A127M, L130F}$ is over 20-fold more sensitive to 3OC10-HSL than wild-type LasR. The L125F substitution appears to be the primary driver of this altered selectivity (*Figure 6B–C* and *Figure 6—figure supplement 2A*). LasR L125 is located in the 3OC12-HSL binding pocket, where it interacts with the end of the signal's acyl tail. The L125F substitution may decrease the size of the binding pocket, improving hydrophobic interactions with shorter acyl chains. All 'MupR-like' LasR

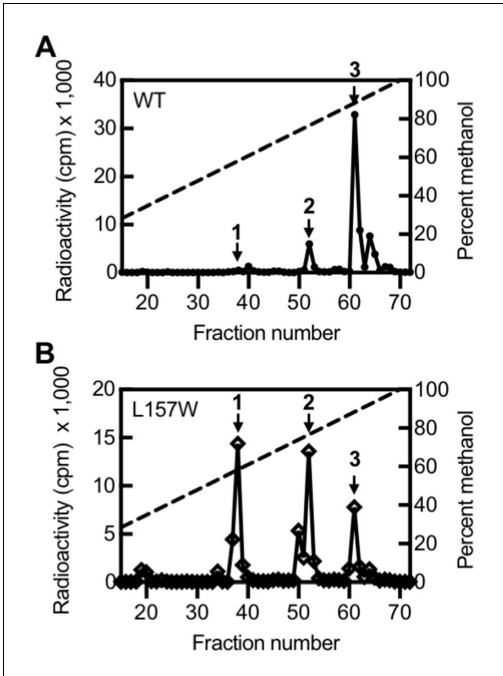

**Figure 5.** Activity of LasI variants. High-performance liquid chromatography (HPLC) analysis of radiolabeled acyl-homoserine lactones (AHLs) extracted from culture fluid of *Pseudomonas aeruginosa* PAO-SC4 harboring (**A**) pJN-RBSlasI[WT] or (**B**) pJN-RBSlasI[L157W]. The horizontal axis denotes the HPLC fraction number (fractions 1–14 are not shown). The methanol gradient is indicated as a dashed line plotted on the right vertical axis. The left vertical axis indicates the amount of radioactivity (counts per minute [cpm]) in each fraction. Data are representative of two (L157W) or three (wild type [WT]) independent experiments. Arrow 1 indicates the fraction in which *N*-3-oxo-octanoyl-L-homoserine lactone (3OC8-HSL) elutes, arrow 2 indicates the fraction in which *N*-3-oxo-decanoyl-L-homoserine lactone (3OC10-HSL) elutes, and arrow 3 indicates the fraction in which *N*-3-oxo-dodecanoyl-L-homoserine lactone (3OC12-HSL) elutes.

The online version of this article includes the following source data and figure supplement(s) for figure 5:

**Figure supplement 1.** Radiotracer assays of LasI activity.

**Figure supplement 1—source data 1.** Phosphor imaging of radio-thin layer chromatography (TLC) shown in *Figure 5—figure supplement 1A*.

**Figure supplement 1—source data 2.** Phosphor imaging of radio-thin layer chromatography (TLC) shown in *Figure 5—figure supplement 1A* with lanes labeled and colors inverted.

**Figure supplement 1—source data 3.** Phosphor imaging of radio-thin layer chromatography (TLC) shown in *Figure 5—figure supplement 1C*.

**Figure supplement 1—source data 4.** Phosphor imaging of radio-thin layer chromatography (TLC) shown in *Figure 5—figure supplement 1C* with lanes labeled and colors inverted.

*Figure 5 continued on next page*

variants responded to 3OC12-HSL with similar sensitivity to wild-type LasR when expressed in *E. coli* (*Figure 6—figure supplement 2B–C*).

While 3OC10-HSL is the native signal for the MupI/R system, the sensitivity of MupR to other signals is unknown. To compare the activity of our LasR variants with that of MupR, we developed a transcriptional reporter of MupR activity in *E. coli*. To do this, we used the promoter of *mupI* to control *gfp* expression in the plasmid pPROBE-GT. LuxR-type receptors often positively regulate their paired synthase gene (*Ng and Bassler, 2009*). Searching the promoter region upstream *mupI*, we identified an inverted repeat centered at −64.5 relative to the start codon of *mupI* that has high similarity (12/20 base pairs) to the LasR-binding site upstream the LasR-regulated gene *rsaL* (*Figure 6—figure supplement 3A*; *Whiteley and Greenberg, 2001*). It is therefore plausible that MupR regulates *mupI*. We created an arabinose-inducible *mupR* expression vector in the plasmid pJN105 and introduced it along with pPROBE-P$_{mupI}$ into *E. coli*. If MupR activates transcription from the *mupI* promoter, this reporter strain should express *gfp* in the presence of 3OC10-HSL. We validated the reporter by measuring GFP fluorescence in the presence or absence of arabinose (to induce *mupR* expression) and/or 3OC10-HSL (*Figure 6—figure supplement 3B*). While there is some leaky expression of *mupR* from the pJN105 vector, indicated by increased fluorescence in the presence of 3OC10-HSL and absence of arabinose, the reporter strain requires both arabinose and 3OC10-HSL for maximal fluorescence. Using this reporter we found that MupR, when expressed in *E. coli*, is equally sensitive to 3OC10-HSL and 3OC12-HSL (*Figure 6—figure supplement 3C*). Thus, our 'MupR-like' LasR variant mimics the activity of MupR in this context.

To confirm our findings in the native context, we engineered mutations into *lasR* on the chromosome of the AHL synthase-null *P. aeruginosa* PAO-SC4. We found that wild-type LasR has minimal 3OC10-HSL activity, whereas the 'MupR-like' LasR[L125F, A127M, L130F] is much more sensitive to 3OC10-HSL and responds to concentrations as low as 1 μM. As in *E. coli*, the 'MupR-like' variant maintains its 3OC12-HSL activity in *P. aeruginosa* (*Figure 6—figure supplement 4A–C*). To determine whether the 'MupR-like' LasR variant stimulates social behaviors, we assessed QS-dependent protease production by plating *P. aeruginosa* on casein agar. In this assay, QS-regulated protease production is required for cell growth, and high levels of protease result in a

*Figure 5 continued*

**Figure supplement 1—source data 5.** Phosphor imaging of radio-thin layer chromatography (TLC) shown in *Figure 5—figure supplement 1D*.

**Figure supplement 1—source data 6.** Phosphor imaging of radio-thin layer chromatography (TLC) shown in *Figure 5—figure supplement 1D* with lanes labeled and colors inverted.

**Figure supplement 1—source data 7.** Phosphor imaging of radio-thin layer chromatography (TLC) shown in *Figure 5—figure supplement 1E*.

zone of clearing around the colony and a white zone of partially degraded casein at the periphery of the clearing (*Chen et al., 2019*). AHL synthase-null *P. aeruginosa* PAO-SC4 grows poorly on casein agar with 3OC10-HSL or with no signal added, but grows well and produces protease in response to 3OC12-HSL. In contrast, *P. aeruginosa* PAO-SC4 LasR[L125F, A127M, L130F] grows on casein agar with either 3OC10-HSL or 3OC12-HSL, indicating that 3OC10-HSL stimulates protease production in the 'MupR-like' variant (*Figure 6D*).

LasI differs from MupI at five high-scoring covariation residues: LasI M125, T145, M152, V159, and N181 (*Figure 7—figure supplement 1A*), the first three of which line the LasI acyl-binding pocket (*Figure 7A*, *Figure 7—figure supplement 1B*). Swapping these three residues for their MupI identities resulted in a synthase that has

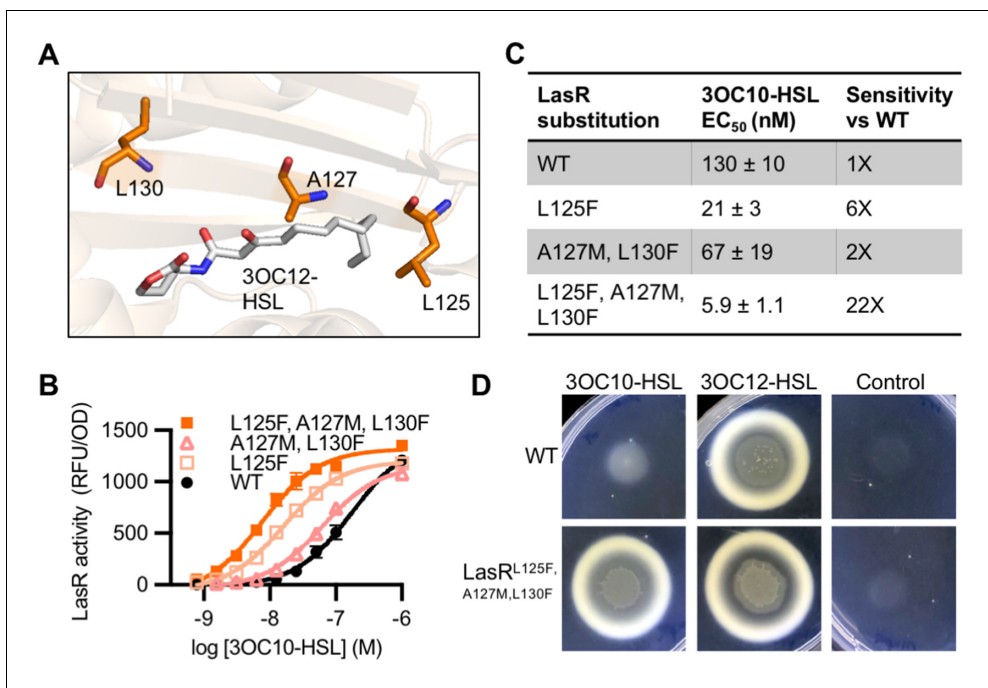

**Figure 6.** Engineering LasR to respond to *N*-3-oxo-decanoyl-L-homoserine lactone (3OC10-HSL). (**A**) Residues altered in LasR shown as orange sticks in the LasR structure (PDB 3IX3). *N*-3-oxo-dodecanoyl-L-homoserine lactone (3OC12-HSL) is shown in gray. (**B**) LasR activity in response to 3OC10-HSL measured in *Escherichia coli* harboring pJNL (wild type [WT], or with indicated amino acid substitutions) and pPROBE-P$_{rsaL}$. Data are the mean and standard deviation of three biological replicates and are representative of three independent experiments. (**C**) Half maximal effective concentration (EC$_{50}$) of 3OC10-HSL for LasR, calculated from three independent experiments (representative data shown in panel **B**). Data are mean and SEM. Sensitivity of mutants compared to LasR[WT] is calculated by dividing the WT EC$_{50}$ by mutant EC$_{50}$. (**D**) Growth of *Pseudomonas aeruginosa* PAO-SC4 LasR[WT] or LasR[L125F, A127M, L130F] on casein agar with 10 μM 3OC10-HSL, 3OC12-HSL, or a vehicle control (DMSO). Data are representative of three independent experiments.

The online version of this article includes the following source data and figure supplement(s) for figure 6:

**Source data 1.** Full images of casein plates shown in *Figure 6D*.
**Figure supplement 1.** Comparison of LasR and MupR protein sequences.
**Figure supplement 2.** 'MupR-like' LasR variant activity.
**Figure supplement 3.** Activity of a MupR transcriptional reporter.
**Figure supplement 4.** Activity 'MupR-like' LasR variants in *Pseudomonas aeruginosa*.

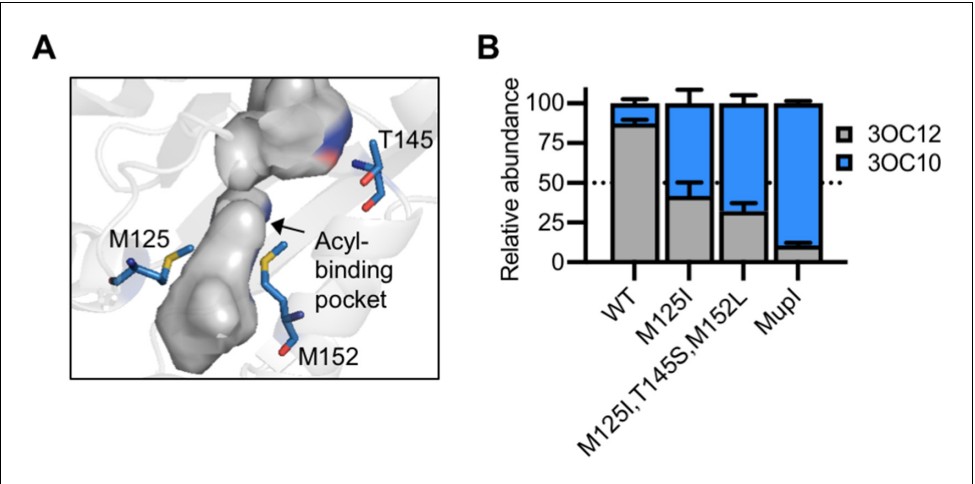

**Figure 7.** Engineering LasI to produce *N*-3-oxo-decanoyl-L-homoserine lactone (3OC10-HSL). (**A**) Residues altered in LasI shown as blue sticks in the LasI structure (PDB 1ROH). (**B**) Relative amount of acyl-homoserine lactones (AHLs) produced by *Pseudomonas aeruginosa* PAO-SC4 harboring pJN-RBSlasI (wild type [WT] or with the indicated amino acid substitutions) or pJN-RBSmupI. Ratios were calculated from high-performance liquid chromatography (HPLC) data (representative data shown in *Figure 7—figure supplement 2A–D*). Bars show mean and standard deviation. The dashed line indicates equal production of 3OC10-HSL and *N*-3-oxo-dodecanoyl-L-homoserine lactone (3OC12-HSL).

The online version of this article includes the following source data and figure supplement(s) for figure 7:

**Figure supplement 1.** Comparison of LasI and MupI protein sequences.

**Figure supplement 2.** 'MupI-like' LasI variant activity.

**Figure supplement 2—source data 1.** Phosphor imaging of radio-thin layer chromatography (TLC) shown in *Figure 7—figure supplement 2E*, which contains images from three TLCs.

---

substantially altered selectivity. LasI$^{M125I, T145S, M152L}$ produces about twofold more 3OC10-HSL than 3OC12-HSL. The M125I substitution alone was sufficient to relax LasI's selectivity, resulting in a synthase that produces roughly equal amounts of 3OC10-HSL and 3OC12-HSL (*Figure 7B*, *Figure 7—figure supplement 2A–C*). LasI M125 is located near the C9 carbon of the acyl chain of a substrate modeled into the LasI structure (*Gould et al., 2006*). Changes to this residue could obstruct binding of substrates with longer acyl chains while increasing affinity for shorter substrates. As a comparison, we measured the activity of *mupI* expressed in *P. aeruginosa*, and found it produces 9:1 3OC10-HSL:3OC12-HSL (*Figure 7B* and *Figure 7—figure supplement 2D*). All single and double 'MupI-like' LasI variants retained AHL synthase activity, but only those that contain the M125I substitution displayed increased 3OC10-HSL production relative to 3OC12-HSL (*Figure 7—figure supplement 2E*).

## Discussion

Despite decades of study, it has been challenging to determine how AHL QS systems distinguish between signals. We hypothesized that we could identify covariation patterns in AHL QS systems and that these patterns would illuminate residues important for signal selectivity. Using a novel application of GREMLIN to analyze the sequences of 3489 unique AHL QS systems, we identified amino acids that strongly covary between AHL synthases and receptors. The top-scoring residues in our analysis cluster near the ligand-binding pockets for both proteins and are more than three times closer to the signal molecule compared to top-scoring residues in a randomized control. We focused our study on *P. aeruginosa* LasI/R. Through targeted alterations in the top-scoring covarying residues, we demonstrate that these amino acids are indeed determinants of signal selectivity. We have thus validated a new application of covariation analysis for proteins that interact indirectly and not through direct binding to one another. This use of covariation analysis for non-physical protein-protein interactions may be useful for other systems in which proteins are connected by small molecules,

for example, metabolic pathways. Additionally, these strong covariation results further support the view that AHL synthases and receptors coevolve.

For both the synthase, LasI, and the receptor, LasR, a single amino acid substitution is sufficient to significantly alter selectivity. Interestingly, our amino acid substitutions also revealed that LasR is not optimized to be as sensitive as possible to its native 3OC12-HSL signal. The increase in sensitivity of specific variants came at the cost of decreased selectivity, which suggests that QS systems may evolve to balance these two properties. Furthermore, increased sensitivity to the native signal may lead to premature activation of the QS regulon, which would likely decrease fitness (*Darch et al., 2012*). The products and behaviors regulated by QS, such as pyocyanin and protease production, are complexly regulated not only by LasI/R, but also by other factors including additional QS receptors (*Brint and Ohman, 1995*; *Ochsner et al., 1994*). The variants generated in our study provide us with the tools to assess the impact of LasI/R sensitivity and selectivity on QS timing and gene expression level and, ultimately, on these more complex social phenotypes.

We also demonstrated that we can use covarying residues to rationally engineer a QS system to produce and respond to a non-native signal. By changing the covarying residues in LasI/R to their MupI/R identities, we improved the sensitivity of LasR to 3OC10-HSL over 20-fold and increased the production of 3OC10-HSL by LasI roughly 15-fold. For both the synthase and receptor, a single amino acid substitution was the primary driver of the altered selectivity. This was surprising given the low sequence identity between LasI/R and MupI/R. These findings suggest new QS systems might evolve with relative ease. Further, the ability to engineer QS selectivity could be beneficial to synthetic biology where AHL signaling is a powerful tool to build biological circuits (*Davis et al., 2015*). Notably, there has not been a previous attempt to simultaneously engineer both an AHL synthase and receptor to use a non-native signal.

Though we were able to substantially increase the 3OC10-HSL activity of LasI/R, our variants retained their native 3OC12-HSL activity. We have thus generated a promiscuous system with broadened selectivity. Similarly, a directed evolution study of the AHL receptor LuxR found that it evolves through promiscuous intermediates (*Collins et al., 2005*). This has also been observed in other systems, such as toxin-antitoxin systems (*Aakre et al., 2015*). Proteins tend to evolve through broadly active intermediates before gaining new specificity. In this way, the system maintains functionality *en route* to altered selectivity. QS systems appear to follow these same trends.

Further work is needed to fully 'rewire' LasI/R to exclusively use a non-native signal and to enable the rational engineering of QS systems to use any signal of choice. One limitation we faced is a lack of close LasI/R homologs with known signals. The identification of a more closely related system to LasI/R may provide a better starting point for engineering altered selectivity. It has been demonstrated that 'supporting' residues, that is, residues within a protein that covary with the selectivity residues, may indirectly impact selectivity by influencing the orientation of selectivity residues (*Aakre et al., 2015*). Thus, given the large differences in sequence identity between the Las and Mup systems, there are likely other residues that must be changed to fully swap selectivity. Supporting residues may be a tractable starting point. Alternatively, our 'Mup-like' variants could serve as the basis for identifying QS proteins with altered selectivity using saturating mutagenesis and/or in vitro evolution. Such studies would illuminate additional determinants of selectivity and potentially uncover 'rules' of signal binding. The variants reported in this study serve as a foundation for future work on signal sensitivity and selectivity and the rational engineering of QS systems.

Collectively, our results provide insight into AHL QS selectivity and will help us predict signal selectivity in newly identified QS systems, in metagenomes, and in naturally occurring QS variants such as those found in clinical isolates. More broadly, we have gained insight into how AHL QS systems evolve and diversify and have validated a new use of covariation analysis for investigating protein-ligand selectivity in coevolving proteins that are connected by a small molecule.

# Materials and methods

**Key resources table**

| Reagent type (species) or resource | Designation | Source or reference | Identifiers | Additional information |
|---|---|---|---|---|

*Continued on next page*

*Continued*

| Reagent type (species) or resource | Designation | Source or reference | Identifiers | Additional information |
|---|---|---|---|---|
| Strain, strain background (*Pseudomonas aeruginosa*) | PAO-SC4 | *Wellington and Greenberg, 2019* | *P. aeruginosa* PAO1 with unmarked deletions of *lasI* and *rhI* | AHL synthase-null mutant |
| Strain, strain background (*Pseudomonas aeruginosa*) | PAO1Δ*rhII* | *Wang et al., 2015* | PAO1 with unmarked in-frame deletion of *rhII* | |
| Strain, strain background (*Pseudomonas aeruginosa*) | PAO1Δ*lasR* | *Wang et al., 2015* | PAO1 with unmarked in-frame deletion of *lasR* | |
| Strain, strain background (*Escherichia coli*) | 5-Alpha | New England Biolabs | *fhuA2 Δ(argF-lacZ)U169 phoA glnV44 Φ80 Δ(lacZ)M15 gyrA96 recA1 relA1 endA1 thi-1 hsdR17* | Chemically competent cells; used for cloning and for LasR and MupR activity reporter strains |
| Strain, strain background (*Escherichia coli*) | S17-1 | *Simon et al., 1983* | *recA pro hsdR RP4-2Tc::Mu-Km::Tn7* | Used for conjugal transfer of plasmid DNA |
| Strain, strain background (*Pseudomonas fluorescens*) | Migula (ATCC 49323) | ATCC | | |
| Antibody | Anti-LasR (Rabbit polyclonal) | Covance; *Gilbert et al., 2009* | | (1:1000) |
| Recombinant DNA reagent | pPROBE-P$_{rsaL}$ | *Wellington and Greenberg, 2019* | *gfp* reporter of LasR activity, Gm$^R$ | |
| Recombinant DNA reagent | pJNL | *Wellington and Greenberg, 2019* | Arabinose-inducible *lasR* expression vector, Ap$^R$ | |
| Recombinant DNA reagent | pJN-empty | *Wellington and Greenberg, 2019* | Arabinose-inducible expression vector with no gene inserted, Ap$^R$ | |
| Recombinant DNA reagent | pJN-lasI | This paper[*] | Arabinose-inducible *lasI* expression vector, Ap$^R$ | Derived from pJN105, see Materials and methods for details |
| Recombinant DNA reagent | pJN-RBSlasI | This paper | pJN-lasI with native *lasI* RBS, Ap$^R$ | Derived from pJN105, see Materials and methods for details |
| Recombinant DNA reagent | pJN-RBSmupI | This paper | Arabinose-inducible *mupI* expression vector, Ap$^R$ | Derived from pJN105, see Materials and methods for details |
| Recombinant DNA reagent | pJN105-mupR | This paper | Arabinose-inducible *mupR* expression vector, Ap$^R$ | Derived from pJN105, see Materials and methods for details |
| Recombinant DNA reagent | pPROBE-P$_{mupI}$ | This paper | *gfp* reporter of MupR activity, Gm$^R$ | pPROBE-GT with the *mupI* promoter extending from −300 to +42, see Materials and methods for details |
| Recombinant DNA reagent | pEXG2 | *Rietsch et al., 2005* | Allelic exchange vector with pBR origin, *sacB*, Gm$^R$ | |
| Recombinant DNA reagent | pEXG2-lasR | Gift from M Kostylev and EP Greenberg | pEXG2 containing *lasR* gene and 500 bp up- and down-stream | |
| Sequence-based reagent | Cloning primers | This paper | | See *Supplementary file 1F* |
| Gene (various) | *lasI, lasR, mupI, mupR* | | | See *Supplementary file 1A* |
| Software, algorithm | GREMLIN | *Ovchinnikov et al., 2014* | Protein covariation analysis algorithm | |
| Chemical compound, drug | $^{14}$C-methionine | American Radiolabeled Chemicals | Methionine, L-[1–14C], 0.1 mCi/mL; SKU ARC 0271 | |

* Please contact the corresponding author (EP Greenberg) to request strains or plasmids created in this study.

## Identification of QS systems

Starting from 24 pairs of manually curated QS synthases and receptors (*Supplementary file 1A*), we searched for homologs in complete bacterial genomes using BLAST (e-value <0.01) (*Altschul et al., 1990*). We filtered the BLAST hits by sequence identity (>30%) to the query and the alignment coverage (query coverage >0.75 and hit coverage >0.75), and the filtered hits were aligned by Clustal Omega (*Sievers and Higgins, 2021*). We selected the LasI/R system from *P. aeruginosa* PAO1 as the target and mapped the multiple sequence alignments (MSA) to the target system. We built sequence profiles from the MSA with HMMER (*Eddy, 2009*) and hmmbuild for the QS synthases and receptors, respectively. The sequence profiles were then used to search against the ENA database (*Amid et al., 2019*) and the IMG/M database (*Chen et al., 2021*) from JGI using HMMER hmmsearch. A total of 149,837 and 5,046,620 homologs were found in these databases for the QS synthase and receptor, respectively. Because the synthases and receptors of the known QS systems frequently locate near each other in the genome, we kept synthase-receptor gene pairs that are separated by no more than two other open reading frames in the genome or contig. A total of 14,980 synthase-receptor gene pairs were identified and they represent 6360 non-identical QS systems. In another attempt, we carried out the same procedure using the LuxI/R system from *V. fischeri* MJ11 as the target system. A similar number of QS systems were identified.

## Identification of covarying residues

We connected the synthase and receptor protein sequences for each QS system we found in the databases and derived the alignments between these QS systems to the target QS system (LasI/R) from the hmmsearch result. We filtered the MSA for synthase-receptor pairs by sequence identity (maximal identify for remaining sequences ≤90%) and gap ratio in each sequence (maximal gap ratio ≤25%). We applied GREMLIN to analyze the covariation in the MSA (*Kamisetty et al., 2013*), and the GREMLIN coefficients were normalized using APC (*Buslje et al., 2009*) as we described previously (*Ovchinnikov et al., 2014*). The GREMLIN coefficients after APC were used as measures for covariation signals between synthase and receptor amino acid residues. As a control, we connected each synthase sequence with a randomly selected receptor sequence and performed the covariation analysis in the same way.

We mapped the top-scoring covarying residues in the LasI/R system onto the crystal structures for each protein. Reported distances between residues and ligands are the shortest distance between any non-hydrogen atoms. For LasR, distances were calculated using PDB 6V7X. For LasI, distances were calculated using a LasI structure with 3-oxo-C12-acyl-phosphopantetheine modeled into the acyl-binding pocket (*Gould et al., 2004*). Reported distances for LasI are between residues and the acyl portion of the modeled substrate.

## Bacterial strains, plasmids, and culture conditions

Bacterial strains and plasmids are listed in the key resources table. Unless otherwise specified, *P. aeruginosa* and *E. coli* were grown in lysogeny broth (LB) (10 g tryptone, 5 g yeast extract, 5 g NaCl per liter) buffered with 50 mM 3-(*N*-morpholino) propanesulfonic acid (MOPS) (pH 7) (LB/MOPS) or on LB agar (LB plus 1.5% Bacto agar) (*Wellington and Greenberg, 2019*). Liquid cultures were grown at 37°C with shaking. For radiotracer TLC experiments, *P. aeruginosa* was grown in Jensen's medium with 0.3% glycerol (*Schaefer et al., 2018*). Casein agar was made using minimal broth plus 1% sodium caseinate (casein broth) and 1.5% agar as previously reported (*Chen et al., 2019*).

For plasmid selection and maintenance, antibiotics were used at the following concentrations: *P. aeruginosa*, 30 μg/mL gentamicin (Gm) and 150 μg/mL carbenicillin (Cb); *E. coli* 10 μg/mL Gm and 100 μg/mL ampicillin (Ap). BD Difco Pseudomonas Isolation Agar was prepared according to manufacturer's directions and supplemented with 100 μg/mL Gm as needed. Where needed for gene expression, L-arabinose (0.4% w/v) was added.

All chemicals and reagents were obtained from commercial sources. AHLs were dissolved either in dimethyl sulfoxide (DMSO) or in ethyl acetate (EtAc) acidified with glacial acetic acid (0.01% v/v). AHLs in DMSO were used at ≤1% of the final culture volume and AHLs dissolved in EtAc were dried

on the bottom of the culture vessel prior to addition of the bacterial culture. DMSO or acidified EtAc was used as a vehicle control where appropriate.

## Plasmid and strain construction

pJN-lasI and pJN-RBSmupI were constructed using *E. coli*-mediated DNA assembly (*Kostylev et al., 2015*). Briefly, for pJN-lasI, *lasI* was amplified from *P. aeruginosa* PAO1 genomic DNA (gDNA) using primers lasI-pJN-F and lasI-pJN-R (*Supplementary file 1F*). pJN105 was amplified using the reverse complement of these primers. The resulting PCR products were treated with the restriction enzyme DpnI to remove the parent template. Both PCR products were then used to transform *E. coli* (NEB 5-alpha). The resulting constructs were confirmed by Sanger sequencing. For pJN-RBSmupI, we began by amplifying *mupI* from *P. fluorescens* Migula (ATCC 49323) gDNA using primers mupI-F and mupI-R. We then used primers mupI-pJN-F and mupI-pJN-R to amplify the *mupI* PCR product and used the reverse complement of these two primers to amplify pJN-RBSlasI. The resulting PCR products were treated the same as for pJN-lasI. We constructed pJN-RBSlasI using restriction digestion. The *lasI* gene, including its upstream ribosomal binding site (RBS), was amplified from *P. aeruginosa* PAO1 gDNA using primers RBS-lasI-F and lasI-pJN-R. pJN-lasI and the RBS-*lasI* PCR product were digested using NheI and SacI, gel or column purified, respectively, ligated by T4 DNA ligase, and transformed into NEB 5-alpha. The resulting constructs were confirmed by Sanger sequencing. Plasmids were introduced into *E. coli* by using heat shock and were introduced into *P. aeruginosa* by electroporation.

Point mutations were introduced to *lasI* and *lasR* on pJN-lasI and JNL or pEXG2-lasR, respectively, using site directed mutagenesis by PCR. Primers were designed to amplify each plasmid while introducing the desired mutation(s). The resulting PCR products were treated with DpnI and were then used to transform NEB 5-alpha. Plasmids from the resulting colonies were screened for the desired mutations by Sanger sequencing. To mutate *lasR* on the *P. aeruginosa* PAO-SC4 chromosome, *E. coli* S17-1 was used to deliver pEXG2-lasR containing various *lasR* mutations to PAO-SC4 via conjugation and potential mutants were isolated as previously described (*Kostylev et al., 2019*). All mutations were confirmed by PCR amplification of *lasR* from the genome followed by Sanger sequencing.

pJN105-mupR was created by amplifying *mupR* from *P. fluorescens* gDNA using the primers mupR-F and mupR-R, which add homology to pJN105, including an RBS. pJN-RBSmupI was amplified with the reverse complement of these two primers to generate the vector backbone with homology to *mupR*. The resulting PCR product was treated with DpnI and then both PCR products were purified using a Monarch PCR and DNA Cleanup Kit (NEB). The two fragments were ligated using Gibson assembly, then transformed into *E. coli* NEB 5-alpha. pPROBE-P$_{mupI}$ was constructed by amplifying the *mupI* promoter (−300 to +42) from *P. fluorescens* gDNA using the primers P$_{mupI}$-F and P$_{mupI}$-R. This PCR product was cleaned up with a Monarch kit, then amplified with P$_{mupI}$-pPR-F and P$_{mupI}$-pPR-R to add homology to the pPROBE-GT vector. pPROBE-GT was amplified with the reverse complement of these two primers. The vector PCR was treated with DpnI, then both PCR products were purified and used to transform *E. coli* NEB 5-alpha. The resulting constructs were confirmed by Sanger sequencing.

## LasR and MupR activity measurements

LasR activity was measured in *E. coli* containing pJNL and pPROBE-P$_{rsaL}$ or in *P. aeruginosa* PAO-SC4 containing pPROBE-P$_{rsaL}$ using previously reported methods (*Wellington and Greenberg, 2019*). Briefly, overnight-grown cultures were diluted 1:100 and grown back to log-phase. For *E. coli*, cultures were grown to an optical density at 600 nm (OD) of 0.3, treated with L-arabinose (0.4%), and incubated with AHLs for 4 hr. MupR activity was measured in *E. coli* harboring pJN105-mupR and pPROBE-P$_{mupI}$ using this same protocol. For *P. aeruginosa*, cultures were grown to an OD between 0.05 and 0.3, were diluted to an OD of 0.01, and then incubated with AHLs for 16–18 hr. LasR activity was measured as GFP fluorescence (excitation 490 nm, emission 520 nm, gain 50) using a Synergy H1 microplate reader (Biotek Instruments). Activity measurements were normalized by dividing by OD$_{600}$ and subtracting background values (fluorescence per OD$_{600}$ for cultures incubated with vehicle control). Half maximal effective concentrations, EC$_{50}$, were calculated using GraphPad Prism.

## LasR immunoblotting

The relative levels of soluble LasR in *P. aeruginosa* PAO-SC4 with unmarked *lasR* mutations were assessed using published methods (*Schuster and Greenberg, 2007*). Briefly, cultures were grown overnight in LB/MOPS, diluted 1:100 in LB/MOPS containing 2 μM 3OC12-HSL, and grown to an $OD_{600}$ of 2. Cells were collected by centrifugation at 4˚C and suspended in LasR purification buffer (25 mM Tris-HCl pH 7.8, 150 mM NaCl, 1 mM ethylenediaminetetraacetic acid, 1 mM dithiothreitol, 0.5% Tween-20, 10% glycerol, 2 μM 3OC12-HSL). The cell suspensions were sonicated and the resulting lysates were subjected to ultracentrifugation at 55,000 rpm for 30 min at 4˚C. Protein concentrations were determined by NanoDrop and normalized samples were separated by SDS-PAGE. The separated proteins were transferred to a PVDF membrane which was treated with polyclonal antibodies against LasR (Covance; 1:1000 dilution). Proteins were detected using a secondary anti-rabbit horseradish peroxidase IgG and chemiluminescent substrate.

## TLC screening for AHLs

Cultures of *P. aeruginosa* PAO1Δ*rhlI* or of *P. aeruginosa* PAO-SC4 with pJN-empty or with wild-type or mutated pJN-lasI were grown overnight in Jensen's medium with 0.3% glycerol. Overnight cultures were used to inoculate fresh medium (1% v/v). When the OD reached 0.5, *lasI* expression was induced with arabinose (0.4%) and 1.1 mL cultures were incubated with 1.1 μCi/mL L-[1-$^{14}$C]-methionine ($^{14}$C-methionine, American Radiolabeled Chemicals) for 90 min (*Schaefer et al., 2018*). Cells were pelleted by centrifugation and 1 mL of supernatant fluid was extracted twice with 2 mL acidified EtAc. The extracts were dried under $N_2$ and resuspended in 15 μL acidified EtAc. 5 μL of each extract was spotted on an aluminum backed C18-W-silica TLC plate (Sorbtech). AHLs were separated using 70% methanol in water, then the TLC plate was dried and exposed to a phosphor screen for at least 16 hr. Phosphor screens were imaged with a Sapphire Biomolecular Imager (Azure Biosystems). To confirm TLC findings, select extracts were dried, suspended in methanol, and analyzed by C18-reverse-phase HPLC using a previously reported method (*Schaefer et al., 2018*).

## HPLC radiotracer assays for LasI activity

For better detection of AHLs, we slightly modified the radiolabeling protocol detailed above, modeling it after a previously published method (*Leadbetter and Greenberg, 2000*). Cultures of *P. aeruginosa* PAO-SC4 with wild-type or mutated pJN-RBSlasI were grown overnight in LB/MOPS. Overnight cultures were used to inoculate 5 mL LB/MOPS (1% v/v). After 2 hr, *lasI* expression was induced with arabinose (0.4%) and cultures were grown to OD 0.7. Cells were centrifuged at 5000 rpm for 10 min, and pellets were suspended in 1.1 mL phosphate buffered saline with 10 mM glucose. After shaking incubation at 37˚C for 10 min, 1.1 μCi $^{14}$C-methionine was added to the cell suspension. Cell suspensions were incubated with radiolabel for 2 hr, after which cells were pelleted by centrifugation and 1 mL supernatant fluid was extracted twice with 2 mL acidified EtAc. Radiolabeled AHLs were dried under $N_2$ and suspended in methanol. One-third of each extract was analyzed by reverse-phase HPLC using a gradient of 10–100% methanol-in-water (*Schaefer et al., 2018*).

## Assessment of protease production on casein agar

Casein agar was used to evaluate protease production as previously described (*Chen et al., 2019*). Cultures were grown in LB/MOPS overnight, diluted to an $OD_{600}$ of 0.1, and then 3 μL was spotted on casein agar plates (60 mm × 15 mm) containing AHLs or DMSO as a vehicle control. Colony growth and casein proteolysis around the colony were assessed after 48 hr at 37˚C.

## Acknowledgements

We thank Mair Churchill for sharing her lab's LasI structure modeled with an acyl substrate. This work was supported by grants NIH R35GM136218 to EPG, Yeast Program Grant 5 P41 GM103533-24 and Washington Research Foundation fellowship to QC, and Helen Hay Whitney Foundation fellowship to SWM.

## Additional information

### Competing interests

E Peter Greenberg: Reviewing editor, *eLife*. The other authors declare that no competing interests exist.

### Funding

| Funder | Grant reference number | Author |
| --- | --- | --- |
| National Institutes of Health | R35GM136218 | Greenberg EP |
| National Institutes of Health | Program Grant P41 GM103533-24 | Cong Q |
| Washington Research Foundation | Postdoctoral Fellowship | Cong Q |
| Helen Hay Whitney Foundation | Postdoctoral Fellowship | Wellington Miranda S |
| Howard Hughes Medical Institute | Investigator Program | Baker D |

The funders had no role in study design, data collection and interpretation, or the decision to submit the work for publication.

### Author contributions

Samantha Wellington Miranda, Conceptualization, Data curation, Funding acquisition, Validation, Investigation, Visualization, Methodology, Writing - original draft, Project administration; Qian Cong, Conceptualization, Data curation, Software, Formal analysis, Funding acquisition, Investigation, Writing - review and editing; Amy L Schaefer, Investigation, Methodology, Writing - review and editing; Emily Kenna MacLeod, Angelina Zimenko, Investigation; David Baker, Conceptualization, Resources, Supervision, Funding acquisition, Writing - review and editing; E Peter Greenberg, Conceptualization, Resources, Supervision, Funding acquisition, Writing - original draft, Writing - review and editing

### Author ORCIDs

Samantha Wellington Miranda  https://orcid.org/0000-0001-5072-2608
Qian Cong  https://orcid.org/0000-0002-8909-0414
David Baker  https://orcid.org/0000-0001-7896-6217
E Peter Greenberg  https://orcid.org/0000-0001-9474-8041

### Decision letter and Author response

Decision letter https://doi.org/10.7554/eLife.69169.sa1
Author response https://doi.org/10.7554/eLife.69169.sa2

## Additional files

### Supplementary files

• Source data 1. Chemiluminescent images for immunoblots, phorphor imaging of radio-thin layer chromatography, and uncropped images of casein plates.

• Supplementary file 1. Supplementary tables containing the following information. (A) Manually curated quorum sensing (QS) synthases and receptors. (B) Select previously reported data for LasR homologs with relevant amino acid substitutions. The native signal for each receptor is indicated in parentheses. (C) Select previously reported data for LasI homologs with relevant amino acid substitutions. The native signal for each synthase is indicated in parentheses. (D) LasR amino acid substitutions evaluated in *Figure 3*. Relative abundance is the frequency of a given amino acid at the indicated position across all LasR homologs. (E) LasI amino acid substitutions evaluated in *Figure 5—*

*figure supplement 1*. Relative abundance is the frequency of a given amino acid at the indicated position across all LasI homologs. (F) Primers used in this study.

• Transparent reporting form

### Data availability

All data generated or analyzed during this study are include in the manuscript and supporting files. Source data files have been provided for the protein sequences analyzed and for Figure 6, Figure 4 - figure supplement 2, Figure 5 - figure supplement 1, and Figure 7 - figure supplement 2.

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
