## [Decision Letter]

**Acceptance summary:**

This paper nicely uses amino-acid covariation analysis to guide mutagenesis work that provides important new insight into the basis of specificity in quorum-sensing systems. The origins of ligand specificity in the LuxI-LuxR family of quorum-sensing systems has been poorly defined and this work now makes major advances in determining the key residues involved. The work lays the foundation for future studies of how selectivity, or a lack thereof, impacts the social behaviors of bacteria.

**Decision letter after peer review:**

Thank you for submitting your article "A covariation analysis reveals elements of selectivity in quorum sensing systems" for consideration by *eLife*. Your article has been reviewed by 3 peer reviewers, including Michael T Laub as the Reviewing Editor and Reviewer #1, and the evaluation has been overseen by Michael Marletta as the Senior Editor. The following individual involved in review of your submission has agreed to reveal their identity: Sampriti Mukherjee (Reviewer #2).

Essential Revisions:

The reviewers have each provided a series of comments and suggestions that the authors should consider and respond to as fully as possible in a revision. From those comments/suggestions, there are 4 of particular note/importance:

1. The authors should revise the manuscript to more carefully note when selectivity has been broadened as opposed to 'rewired'. And, similarly, claims about changes in selectivity should be more precisely stated and bolstered with descriptions of which data support the claims made.

2. Two reviewers felt that the paper was sometimes written for quorum-sensing experts. Efforts to make the paper, both text and figures, more accessible to a wide audience are required.

3. The authors should show, for at least some of their mutants, whether the solubility and/or stability has changed and accounts for the observed changes in sensitivity and/or selectivity.

4. The work relies solely on reporter assays and does not demonstrate changes in selectivity or sensitivity in a native quorum-sensing context. Although not essential to do, if the data are readily obtained, the reviewers felt that it could further strengthen the paper.

*Reviewer #2 (Recommendations for the authors):*

Could the authors elaborate on (a) why the highest GREMLIN scores for top covarying residues of LasI/R pair were so low (max being 0.109), and (b) the basis for choosing the MupI/R system as the target.

Figure-1: In addition to the current panels, it would be helpful to include a schematic for the GREMLIN APC pipeline so that the readers clearly understand the workflow.

Figure-4: LasIL157W produces 3OC8-HSL. Is there a LasR variant based on the GREMLIN approach that could detect 3OC8-HSL?

Figure-5a: A zoomed out view, i.e., annotating the residues in LasR and LasI AHL binding pockets in the context of the entire protein structure would be helpful for the readers.

Figure-5b: How does the "MupR"-like LasR variant control quorum sensing in P. aeruginosa? 5b shows data only for *E. coli* reporter assay. At least the reporter assay data for these LasR variants in P. aeruginosa should be included.

Line 142: "Several mutants produced little or no detectable AHLs"…are these LasI mutants stably expressed?

Line 145: "Based on our TLC results we selected one mutant"…could the authors clarify the basis for selecting LasIL157W while leaving out T142A or T145A?

*Reviewer #3 (Recommendations for the authors):*

I can offer only one substantive suggestion for improvement of the science, which may well fall into the realm of future investigation. The approach used in the manuscript of examining co-variation between synthase and receptor pairs makes sense and was obviously productive. With the I and R sequence databases already in hand, it presumably would be simple (and likely informative) to also examine covariation within synthases and within receptors. I expect that the positions identified as important via covariation between synthases and receptors will turn out to be linked to one another by covariation within synthases or receptors. Furthermore, when making multiple substitutions to engineer altered specificity of synthases or receptors, as was done in Figure 5 and Supplementary Figures 5 and 6, the network of covariation within a family seems likely to identify the most important subset of residues to change. Finally, it seems likely that the sequence space defined by naturally occurring combinations of amino acids at covarying positions within a protein family will be limited and likely predictive of specificity. This approach would address the challenge raised by the authors in lines 211-215, but offers a different solution than the authors suggest on lines 215-218.

---

## [Author Response]

Essential Revisions:The reviewers have each provided a series of comments and suggestions that the authors should consider and respond to as fully as possible in a revision. From those comments/suggestions, there are 4 of particular note/importance:1. The authors should revise the manuscript to more carefully note when selectivity has been broadened as opposed to 'rewired'. And, similarly, claims about changes in selectivity should be more precisely stated and bolstered with descriptions of which data support the claims made.

We did not intend to overstate our findings and have revised the text to avoid confusion on this matter as well as to provide more descriptions of the changes in selectivity we observed.

2. Two reviewers felt that the paper was sometimes written for quorum-sensing experts. Efforts to make the paper, both text and figures, more accessible to a wide audience are required.

We have added text discussing various aspects of quorum sensing and our methods (e.g. lines 164-168, 186-219). We have also added a figure showing the structures of the AHL signals in our study along with the names by which we refer to them in the manuscript (Figure 3—figure supplement 1) and have made reviewer recommended changes to several figures (detailed in the response below). We hope this makes the paper more accessible and welcome additional reviewer suggestions.

3. The authors should show, for at least some of their mutants, whether the solubility and/or stability has changed and accounts for the observed changes in sensitivity and/or selectivity.

The abundance and stability of LuxR-type receptors is intertwined with signal binding. A key component of the natural regulation of AHL QS systems is that receptors, including LasR, are unstable/insoluble in the absence of bound signal. Thus, changes in affinity for a signal often lead to changes in the amount of soluble receptor present in a cell. Mutant LasR sensitivity and selectivity could be altered for many reasons including changed signal affinity, protein solubility, or, likely, both, and cause and effect is difficult to determine. Small differences in LasR level could conceivably correspond to the differences in 3OC12 sensitivity we see, even if these differences were not detected by immunoblotting. We have added a discussion of the link between LasR solubility and signal binding and the factors that may account for changes in LasI/R sensitivity and selectivity to the text (lines 186-219, 252-254).

We have also performed additional experiments to measure LasR solubility in some of the mutants (new Figure 4—figure supplement 2). These studies show that there is not a substantial difference in the level of soluble LasR between the variants and wild type (lines 219-224). Unfortunately we do not have a LasI antibody available for similar studies on synthase variants.

4. The work relies solely on reporter assays and does not demonstrate changes in selectivity or sensitivity in a native quorum-sensing context. Although not essential to do, if the data are readily obtained, the reviewers felt that it could further strengthen the paper.

We chose to use reporter assays for our study because they are the best way to directly measure the primary activity of LasR, which is as a transcriptional activator. This is the most appropriate method by which to draw conclusions about LasR sensitivity and selectivity in a whole cell, and by studying LasI/R in *P. aeruginosa*, we have measured their activity in the native context. Whether changes in LasI/R sensitivity and selectivity affect sociality is a very intriguing, but separate, question. The products regulated by quorum sensing, such as pyocyanin and protease, are complexly regulated not only by LasR, but also by other factors including additional quorum sensing receptors. Changes in the sensitivity and selectivity of LasI/R could impact the timing and expression level of these products, and, ultimately, strain fitness in various environments. This is a subject we are excited to research in the future.

As a step toward these future research goals we have performed additional experiments and demonstrate that “MupR-like” LasR promotes a social phenotype in *P. aeruginosa* (production of secreted protease) in response to the newly detected signal 3OC10-HSL (new Figure 6D).

Reviewer #2 (Recommendations for the authors):Could the authors elaborate on (a) why the highest GREMLIN scores for top covarying residues of LasI/R pair were so low (max being 0.109), and (b) the basis for choosing the MupI/R system as the target.

The scores we reported are not original GREMLIN scores; they are the GREMLIN scores with Average Product Correction, which is a common practice to improve the performance of GREMLIN prediction. The original GREMLIN scores for top-ranking residue pairs between the synthases and the receptors are about 0.4, which is also not a high value. From what we have observed, residues at the interface of directly interacting proteins generally have lower GREMLIN scores than contacts within the same protein, and transiently interacting proteins will have lower scores than obligate interactions. This is expected because the strength of the coevolution signal should correlate with how important it is to maintain a specific contact. Because the synthases and receptors do not directly interact, it is not a surprise to us that they have GREMLIN scores lower than we normally see in proteins that directly interact.

The lower GREMLIN score for indirectly interacting proteins does impose a difficulty in deciding the threshold for confident prediction. Therefore, in this particular study, we used a randomized control to help us determine which scores could be considered strong: those that are higher than the randomized control. We have added new Figure 2B to show a comparison of the GREMLIN analysis (with APC) for synthases and receptors compared to the randomized control. This figure also highlights the threshold for scores that are higher than the randomized control.

MupI/R was chosen as the target sequence because it is among the closest LasI/R homologs (by sequence identity) that uses a signal other than 3OC12-HSL. We have added this explanation to the text (271-272).

Figure-1: In addition to the current panels, it would be helpful to include a schematic for the GREMLIN APC pipeline so that the readers clearly understand the workflow.

We have added our workflow as suggested. This is new Figure 2—figure supplement 1.

Figure-4: LasIL157W produces 3OC8-HSL. Is there a LasR variant based on the GREMLIN approach that could detect 3OC8-HSL?

Several of our mutants respond more strongly to 3OC8-HSL than WT (e.g. A127L and A127M). While we can rationalize some outcomes based on crystal structures, we are not yet at a point where we can predict signal from sequence or use GREMLIN to rationally engineer any desired response. This is a goal and a focus for future research.

Figure-5a: A zoomed out view, i.e., annotating the residues in LasR and LasI AHL binding pockets in the context of the entire protein structure would be helpful for the readers.

We have added these images to the supplemental figures dealing with these variants (new Figure 6—figure supplement 1B and Figure 7—figure supplement 1B).

Figure-5b: How does the "MupR"-like LasR variant control quorum sensing in P. aeruginosa? 5b shows data only for *E. coli* reporter assay. At least the reporter assay data for these LasR variants in P. aeruginosa should be included.

As in *E. coli*, “MupR-like” LasR in P. aeruginosa is much more responsive to 3OC10-HSL than is WT LasR. We have added these data as Figure 6—figure supplement 4. Additionally, “MupR-like” LasR produces protease in response to 3OC10-HSL while WT LasR does not (new Figure 6D). A description of these new experiments is provided in the main text on lines 330-342.

Line 142: "Several mutants produced little or no detectable AHLs"…are these LasI mutants stably expressed?

Unfortunately, we do not have an antibody for LasI and do not know if the variants are stably produced. The reason behind the altered activity of particular mutants is not a focus of this study; that selectivity and the rate of synthesis are changed verifies that covariation analysis works to identify selectivity residues in QS systems, regardless of the mechanisms behind those changes. We have added text to acknowledge the possibility that changes in LasI activity could be due to changes in protein stability (lines 252-254).

Line 145: "Based on our TLC results we selected one mutant"…could the authors clarify the basis for selecting LasIL157W while leaving out T142A or T145A?

We have added text to elaborate on this choice (lines 256-258). LasI^L157W^ showed altered selectivity by TLC and we wanted to know which signals were being produced and in what ratio. This mutant was sufficient to demonstrate that 1) TLC is an appropriate screening method and 2) changes to LasI covarying residues can alter selectivity. Additionally, the COVID pandemic has limited our ability to conduct more HPLC experiments because they are difficult to execute while socially distanced and the ^14^C-methionine substrate necessary for these studies is backordered.

Reviewer #3 (Recommendations for the authors):I can offer only one substantive suggestion for improvement of the science, which may well fall into the realm of future investigation. The approach used in the manuscript of examining co-variation between synthase and receptor pairs makes sense and was obviously productive. With the I and R sequence databases already in hand, it presumably would be simple (and likely informative) to also examine covariation within synthases and within receptors. I expect that the positions identified as important via covariation between synthases and receptors will turn out to be linked to one another by covariation within synthases or receptors. Furthermore, when making multiple substitutions to engineer altered specificity of synthases or receptors, as was done in Figure 5 and Supplementary Figures 5 and 6, the network of covariation within a family seems likely to identify the most important subset of residues to change. Finally, it seems likely that the sequence space defined by naturally occurring combinations of amino acids at covarying positions within a protein family will be limited and likely predictive of specificity. This approach would address the challenge raised by the authors in lines 211-215, but offers a different solution than the authors suggest on lines 215-218.

We thank the reviewer for their engaging summary/public review and their suggestions for our research. We sincerely appreciate the attention to detail and the insight provided by the reviewer.

Though we have not presented the data in the manuscript, we have indeed identified residues that covary within synthases and within receptors. We believe this might point us toward residues that interact with the selectivity residues and play a role in properly orienting them. These amino acids could serve as additional targets of substitutions for engineering. Additionally, some of the selectivity residues do covary with one another in the analysis within synthase and within receptors. At a first pass, these do not form networks that would have told us the most important selectivity residues from the start. We think this is because the strongest coevolution is between residues that directly interact with each other: they need to change accordingly to ensure the stability of proteins. The residues that are important for the cooperation between synthases and receptors will coevolve but to a lower extent.